# Comprehensive multi-omics analysis reveals the prognostic and immune regulatory characteristics of the PTPN family in osteosarcoma

Changhai Long[1☯], Biao Ma[1☯], Xingshun Zhong[1], Mingzhi Zou[2*], Kai Li[1*], Sijing Liu[1,3*]

1 Department of Orthopaedics, Guangdong Medical University Affiliated Second Hospital, Zhanjiang, Guangdong, China, 2 Department of Pharmacy, Guangdong Medical University Affiliated Second Hospital, Zhanjiang, Guangdong, China, 3 Department of Joint Surgery II, Maoming Hospital of Guangzhou University of Chinese Medicine, Maoming, Guangdong, China

☯ These authors also contributed equally to this work.
* lsjlm808@163.com (SJL); kai257257@outlook.com (KL); zmz18966@163.com (MZZ)

## Abstract

Osteosarcoma is a highly aggressive bone tumor that primarily affects adolescents and young adults, posing significant challenges in therapeutic efficacy, prognostic assessment, and treatment strategies. This study investigates the oncogenic and immune regulatory roles of the PTPN family in osteosarcoma using a comprehensive multi-omics approach. We utilized transcriptomic data, single-cell RNA sequencing (scRNA-seq), and clinical information obtained from publicly available databases. Dimensionality reduction and clustering techniques were employed to subclassify immune cells and analyze the tumor microenvironment characteristics. We identified prognostic genes associated with the PTPN family and stratified osteosarcoma cases into distinct molecular subtypes using consensus clustering. A random forest model revealed that the PTPN family has a significant impact on prognosis and modulates key oncogenic pathways. Furthermore, we analyzed the role of the PTPN family in regulating immune cells and selected PTPN23 for experimental validation. This research not only enhances prognostic assessments in osteosarcoma but also establishes a foundation for personalized therapeutic interventions.

## Introduction

Osteosarcoma (OS) is a highly aggressive bone tumor that primarily affects children and adolescents, with a higher incidence observed in individuals aged 10–30 [1–3]. This tumor typically occurs in rapidly growing skeletal areas, such as the metaphyseal regions of long bones in the limbs, with common sites including the distal femur and proximal tibia [4]. Furthermore, the 5-year survival rate for osteosarcoma has stagnated for decades, with poorer outcomes observed in males, American Indian and Alaska Native populations,

# PLOS One

**Data availability statement:** "All relevant data are within the paper and its Supporting Information files".

**Funding:** Zhanjiang Science and Technology Bureau (Project No. 2022A01144 to S.L.), Guangdong Provincial Hospital of Traditional Chinese Medicine (Project No. 20232212 to S.L.), Guangdong Medical Research Foundation (Acceptance No. 20201116152147949 to S.L.), and Zhanjiang Science and Technology Bureau (Project No. 2021A05097 to M.Z.).

**Competing interests:** The authors have declared that no competing interests exist.

elderly patients, and those with metastases, axial tumors, or recurrent cases [5]. Currently, the treatment paradigm for osteosarcoma is relatively well-established, typically involving neoadjuvant chemotherapy, surgical resection, and subsequent adjuvant chemotherapy [6,7]. This comprehensive treatment plan has significantly improved overall survival rates. However, for advanced or metastatic osteosarcoma, the existing treatment options remain limited, and the outcomes are often suboptimal [8]. Therefore, there is an urgent need to develop new systemic therapeutic strategies to improve patient outcomes.

In recent years, immunotherapy has made remarkable progress in the treatment of various malignant tumors and has gradually shown potential in the management of osteosarcoma [9,10]. Some studies suggest that osteosarcoma may originate from mesenchymal stem cells (MSCs), which possess the ability to differentiate into various cell types, including osteoblasts [11,12]. Additionally, tumor neoantigens derived from osteosarcoma cells, along with components of the immune microenvironment, such as tumor-associated macrophages and T cells, represent potential targets for novel immunotherapeutic strategies [13]. In osteosarcoma, macrophages and T cells play crucial roles, with $C1Q^+$ tumor-associated macrophages (TAMs) acting as key mediators of anti-tumor immunity [14]. T cells are essential in osteosarcoma, and blocking TIGIT enhances the cytotoxic effects of $CD3^+$ T cells within the $TIGIT^+$ cell-rich microenvironment [9].

The PTPN family of proteins comprises non-receptor tyrosine phosphatases that play significant roles in immune regulation, cell proliferation, and differentiation [15]. In the context of tumor immunity, PTPN family members (PTPN6, PTPN3, PTPN2, PTPN1) shape the tumor microenvironment by regulating the development and function of immune cells, thereby influencing tumor progression and the efficacy of immunotherapy [16–18]. These phosphatases are critical for modulating immune checkpoint pathways, such as PD-1 and BTLA, which play a key role immune evasion by tumors [19]. Inhibiting the PTPN family enhances intratumoral IFN-γ signaling, increases chemokine release, and facilitates the recruitment of cytotoxic T cells. However, their specific roles in osteosarcoma progression remain unclear, with limited analysis of their immune regulatory functions and prognostic correlations. This study aims to investigate the PTPN family's role in osteosarcoma and its relationship with the immune microenvironment to inform future therapeutic strategies.

## Methods

### Downloading osteosarcoma datasets and collecting PTPN family genes

We obtained RNA expression and clinical data from 84 osteosarcoma samples through UCSC XENA. Additionally we collected single-cell RNA sequencing data (GSE162454) from GEO for six patients. Our focus was on PTPN family genes (PTPN1 to PTPN23), and we downloaded TCGA Pan-Cancer and GTEx data from UCSC XENA.

### Consensus clustering analysis of PTPN family gene expression across different cancer subtypes

We constructed an expression matrix of the PTPN family in osteosarcoma samples and performed consensus unsupervised clustering analysis using the

"ConsensusClusterPlus" tool in R. The assessment criteria included a steadily enhanced cumulative distribution function (CDF) curve, adherence to sample size thresholds for each category, and increased intra-group correlation coupled with decreased inter-group correlation following clustering. Additionally, principal component analysis (PCA) was employed to evaluate subgroup differences, revealing distinct characteristics. Kaplan-Meier (KM) curves were generated to visually illustrate survival differences among the subgroups.

## Functional and pathway enrichment analysis

We identified differentially expressed genes (DEGs) from 84 osteosarcoma samples in clusters 1 and 2 using the limma package (FDR < 0.05, |log2FC| > 1). We performed Hallmark gene set and KEGG analyses on the DEGs using Gene Set Enrichment Analysis (GSEA) within the clusterProfiler package, visualizing the results with heatmaps and enrichment plots. To assess differences in immune infiltration, we utilized the IOBR package to generate box plots based on immune scores (IPS), TIMER, ESTIMATE, and MCPcounter [20–22]. Additionally, we examined correlations between PTPN family gene expression and immune cell infiltration scores, focusing specifically on the relationship between PTPN23 enrichment scores and MCPcounter, which was illustrated through an enrichment plot.

## Data processing and analysis of osteosarcoma single-cell samples

We preprocessed and standardized single-cell RNA sequencing (scRNA-Seq) data from six osteosarcoma samples using the "Seurat" R package. Data filtering excluded genes expressed in fewer than three cells, cells with gene counts below 500 or above 6,000, low-expression genes, and cells with more than 10% mitochondrial content. The data were normalized using the LogNormalize method, followed by principal component analysis (PCA) and uniform manifold approximation and projection (UMAP), employing the first 30 principal components for dimensionality reduction. Clustering was optimized by adjusting the resolution of Seurat's 'FindClusters' function. Marker genes were used to validate annotations, and t-SNE provided two-dimensional visualizations, while heatmaps illustrated clustering results. We conducted AUCell enrichment analysis using PTPN family genes (PTPN1-PTPN23), highlighting their significance in both the Active and Inactive groups [23]. For differential expression analysis, we utilized the FindAllMarkers function, applying p-value and minimum expression thresholds to filter for significant genes. Multi-group volcano plots were generated to display these genes, while stacked charts illustrated proportional changes in cell types between Active and Inactive groups. Additionally, we employed the AggregateExpression function to create pseudobulks from the single-cell data, followed by differential analysis using DESeq2 and Gene Set Variation Analysis (GSVA) with the Hallmark dataset; the results were presented in scatter plots.

## Transcription factor regulatory activity

In this study, we performed the analysis in the PySCENIC Conda environment to ensure consistency with its dependencies and software versions, adhering to the default parameter settings throughout [24]. We performed enhancer activity analysis using PySCENIC to predict transcription factor activity. Utilizing the CisTARGET database, we analyzed DNA sequences located within 500 base pairs upstream of the transcription start site (TSS) and within surrounding regions (5Kb and 10Kb) to identify binding sites, thereby constructing a regulatory network that links transcription factors to their target genes.

## Analysis of t cell and macrophage type changes and regulatory mechanisms in different subgroups

This study utilized the ProjectSVR method for cell subgroup annotation in osteosarcoma samples to ensure accurate mapping, inferring cell types in unknown samples based on the expression patterns of known cell marker genes [25]. We employed Zeming Zhang's pan-cancer annotation model to categorize CD4+ and CD8+ T cells into subsets. Differential

expression analysis, conducted using the SCP R package, identified significant gene differences between clusters 1 and 2, which were visualized in heatmaps. Pathway enrichment analysis employed Gene Ontology (GO) and KEGG tools, and these results were also represented as heatmaps. Additionally, pseudotime analysis using Monocle2 inferred cell state transitions, which were further validated by CytoTRACE, allowing us to explore changes in PTPN family gene expression through scatter plots. Finally, key transcription factors were predicted, and a regulatory network was constructed using regulon analysis tools.

## CellChat cell communication analysis

We constructed a CellChat object using the preprocessed expression matrix and divided the cells into two groups: Active and Inactive. Utilizing CellChat's built-in ligand-receptor database, we extracted potential interactions between these two groups. Our analysis focused on the interactions among multiple ligand-receptor pairs, calculating the interaction strength between different cell types, and thoroughly exploring the significant differences in these signaling pathways between the Active and Inactive groups.

## Pan-cancer analysis of PTPN23

We collected and preprocessed gene expression data and clinical information for TCGA pan-cancer samples from the GDC Data Portal, including normal controls from GTEx. After performing quality control, we compared PTPN23 expression across various cancer types and stages. Prognostic analysis using the Cox model evaluated the relationship between PTPN23 expression and patient outcomes, which was visualized in a forest plot. Immune cell infiltration was analyzed using MCPcounter to correlate PTPN23 expression with immune levels. Additionally, Pathway enrichment analysis via GSEA assessed correlations between PTPN23 expression and Hallmark pathway scores, suggesting its potential role in cancer.

## Cell culture

The 143B and SJSA-1 cell lines were acquired from Pricella (Wuhan, China) and cultured in DMEM that was supplemented with 10% fetal bovine serum and antibiotics. The cells were maintained in a humidified environment at 37°C with 5% $CO_2$. Subculturing was carried out every 2–3 days, with the schedule adjusted according to the growth conditions.

## siRNA Transfection

To investigate the role of PTPN23 in osteosarcoma, we obtained siRNA targeting PTPN23 and a control from Guangzhou Ruibo Biotechnology. Transfection was performed on the 143B and SJSA-1 cell lines using Lipo8000 reagent. Forty-eight hours after transfection, samples were collected for protein quantification to evaluate the silencing effects. The siRNA sequences used were: Sense: 5'- GGACUGGAAGAAACUUGUGCA-3' and Antisense: 5'- CACAAGUUUCUUCCAGU CCUU-3', which specifically target PTPN23 mRNA, while the control was included to assess non-specific effects.

## CCK8 and plate clonogenic cell viability assay

After transfection, we employed the CCK-8 reagent (Beyotime, Shanghai) to assess cell proliferation. Transfected cells were incubated at 37°C, and the absorbance was measured at 450 nm at 24, 48, and 72 hours. For clonogenic assessment, transfected cells were seeded in six-well plates and cultured for 12 days, with regular medium changes. Following this period, cells were fixed with paraformaldehyde and stained with crystal violet for colony counting, which was documented using a digital camera.

## EdU cell proliferation assay

Cells were seeded in 96-well plates at an approximate density of 5000 cells per well and cultured until they reached 70–80% confluence. Following this, EdU was added for 2–4 hours to facilitate incorporation into newly synthesized DNA.

The cells were then fixed with 4% paraformaldehyde and stained using the EdU detection reagent. Fluorescence signals were evaluated using either a microscope or a microplate reader to assess cell proliferation. Untreated cells acted as negative controls, and we ensured that at least three biological replicates were included for each experimental condition to strengthen the reliability and reproducibility of the results.

### Cell cycle (PI)

In this experiment, we performed cell cycle analysis using flow cytometry to assess the impact of si-PTPN23 on the proliferation of 143B and SJSA-1 osteosarcoma cell lines. After treatment, the cells (si-NC and si-PTPN23) were collected and fixed with 70% cold ethanol for 12 hours. Subsequently, the cells were washed with PBS and resuspended in RNase A solution to remove RNA. Following this, PI (propidium iodide) staining was performed to label the DNA. The stained cells were then analyzed using a flow cytometer. The proportions of cells in different phases of the cell cycle (G0/G1, S, G2/M) were analyzed using FlowJo software, and the results were presented as bar charts using R language.

### Western blotting

SDS-PAGE (Epizyme, Shanghai) was used to separate proteins extracted from cell samples. Following separation, the proteins were transferred to a 0.2-micron PVDF membrane, which was subsequently blocked to prevent non-specific binding. The membrane was then incubated overnight at 4°C with the primary antibody. On the following day, a secondary antibody was applied, and the membrane was washed three times with TBST to remove any unbound antibodies. Finally, ECL substrate (Biosharp, Beijing) was added to the membrane, and luminescent signals were detected to quantify the expression levels of the target proteins.

### Statistical analysis

We employed the Wilcoxon rank-sum test or Kruskal-Wallis test to determine differences between groups. A two-sided P-value of less than 0.05 was considered statistically significant.

## Results

### PTPN family discrimination of osteosarcoma sample subgroups

After downloading osteosarcoma expression and clinical data from the UCSC Xena database, we performed consensus clustering (using ConsensusClusterPlus) on TARGET-OS samples based on the PTPN family gene expression (Fig 1A). This analysis divided the samples into two groups: Cluster 1 and Cluster 2. Kaplan-Meier survival curves indicated a significant difference in survival rates between these subtypes (P = 0.021, Fig 1C). Principal component analysis (PCA) further confirmed the differences between the two PTPN subtypes (Fig 1B).

### Gene set variation analysis (GSVA) between two PTPN subtypes

To explore the regulatory role of the PTPN family in osteosarcoma, we conducted Gene Set Variation Analysis (GSVA) on expression profiles. The heatmap revealed significant differences in pathway enrichment, indicating functional diversity among molecular subgroups. Notably, PTPN subtypes exhibited significant pathway inhibition in Hallmark analysis, correlating with poor prognosis, which was also validated in KEGG analysis (Figs 1D and 1E). Hallmark pathway scoring revealed significant enrichments in inflammatory pathways, along with associations with P53 signaling, apoptosis, and G2M cell cycle regulation, particularly highlighting the IL6-STAT3 pathway (Fig 1F). The results of the KEGG analysis revealed involvement in key biological pathways such as WNT signaling, ECM remodeling, VEGF-mediated angiogenesis, cell cycle regulation, and apoptosis (Fig 1G). We employed random forest analysis to find out the impact of the PTPN family on Cluster1 and Cluster2, with PTPN6 and PTPN23 playing significant roles (Fig 1H). Based on the findings, the PTPN

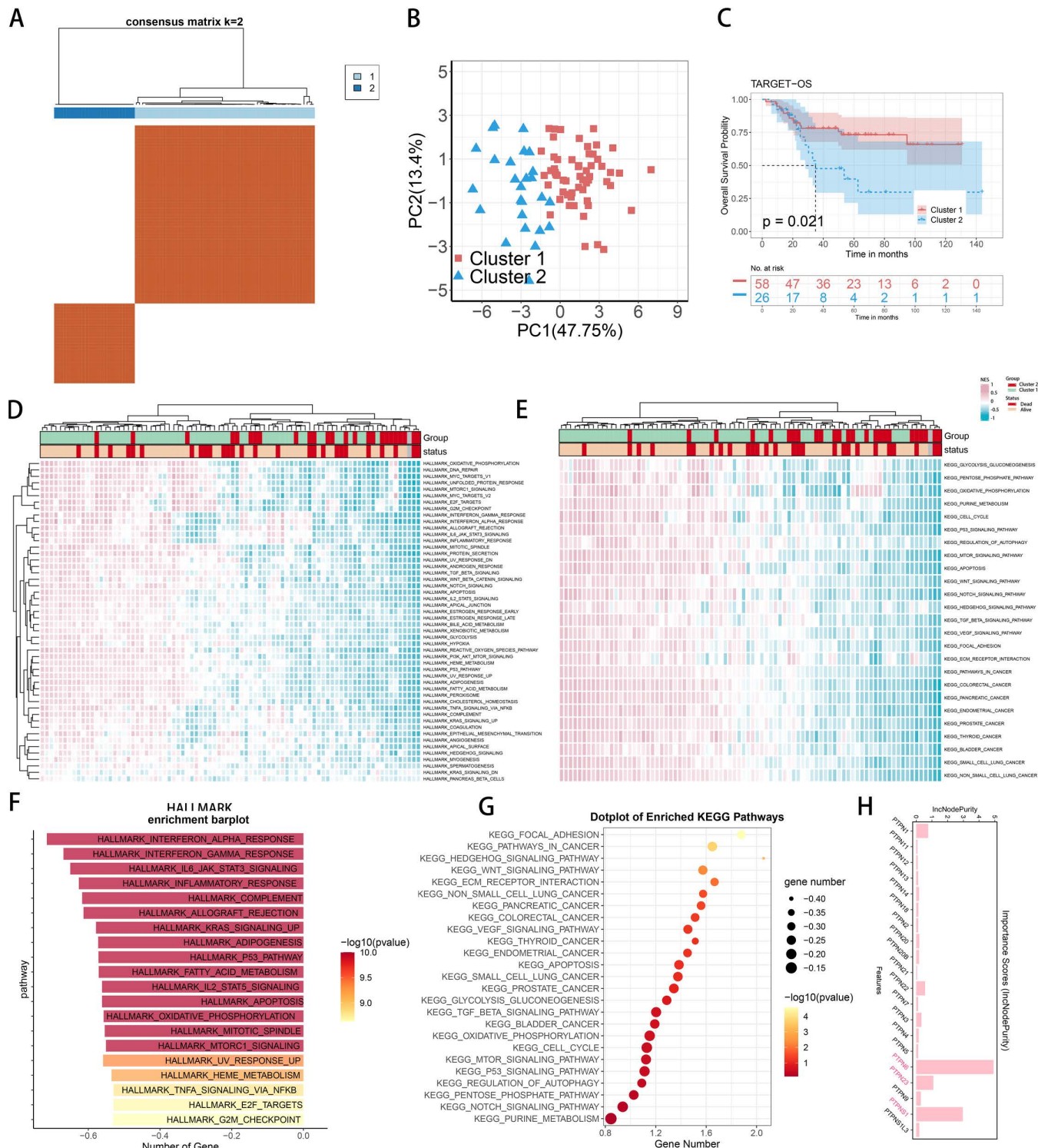

**Fig 1. Identifies two clusters with distinct survival outcomes, showing that upregulated pathways in Cluster 2 are linked to poorer prognosis.** (A) Consensus matrix (k = 2) shows two clusters. (B) Scatter plot differentiates Cluster 1 (red) and Cluster 2 (blue). (C) Kaplan-Meier curves indicate Cluster 2 has poorer prognosis (p = 0.021). (D) GSEA heatmap for Hallmark gene sets correlates Cluster 2 with worse survival. (E) GSEA heatmap for KEGG gene sets shows enriched pathways. (F) Bar chart indicates higher Hallmark pathway enrichment in Cluster 2. (G) Scatter plot highlights KEGG pathway enrichment in Cluster 2 and survival outcomes. (H) The histogram shows the impact of the PTPN family on Cluster1 and Cluster2 through random forest.

family, particularly PTPN6 and PTPN23, appears to play a crucial role in the regulation of osteosarcoma. Their involvement in inflammatory responses and tumor-related pathways may contribute to the development and invasiveness of this cancer type, forming a distinct molecular subgroup associated with poor prognosis. This indicates a potential avenue for targeted therapeutic strategies and biomarkers in the treatment and prognosis assessment of osteosarcoma.

## PTPN family regulation of osteosarcoma immunosuppression and its impact on prognosis

We performed a comprehensive immune infiltration analysis on TARGET-OS samples utilizing methods such as IPS, TIMER, ESTIMATE, and MCPcounter (Fig 2B to 2D). The samples were categorized into Cluster 1 and Cluster 2, revealing a significant reduction in immune cell expression in Cluster 2, suggesting a more pronounced state of immunosuppression. This finding, together with the significantly poorer prognosis for patients in Cluster 2, reinforces the hypothesis that the PTPN family plays a crucial role in regulating immunosuppression. Furthermore, our analysis indicated that the immune regulatory function of the PTPN family is dependent on specific immune mediators, including SC (cytokines or signaling pathways), CP (complement pathways), and AZ (immune-related factors) (Fig 2A). These mediators are vital for immune responses, underscoring the potential role of the PTPN family in immunosuppression and tumor immune evasion.

## Correlation Analysis of the PTPN Family with Immune Cell Infiltration in Osteosarcoma

To investigate the role of the PTPN family in regulating immune cells, we performed immune infiltration analysis using CIBERSORT and Quantiseq methods, followed by correlation analysis with PTPN gene expression (Fig 2E). We discovered that most genes in the PTPN family showed a negative correlation with T cells and macrophages. Notably, PTPN23 exhibited significant correlations with key immune cell types, particularly macrophages, CD4$^+$ T cells, and CD8$^+$ T cells, highlighting its important role in regulating these immune cells. Further MCPcounter analysis revealed a strong correlation between PTPN23 and the expression levels of monocytes and T cells, especially CD8$^+$ T cells (Fig 2F). This correlation underscores the significant regulatory role of PTPN23 in the osteosarcoma immune microenvironment and highlights its potential involvement in the functional regulation of macrophages and T cells. Previous studies have demonstrated that the absence of PTPN1 and PTPN2 in dendritic cells results in increased production of IL-12 and IFN-γ, thereby enhancing the antitumor activity of T cells [26,27]. Similarly, the absence of PTPN2 in T cells can enhance the infiltration and cytotoxicity of CD8$^+$ T cells [28–31]. Conversely, the knockout of PTPN6 significantly enhances the tumor-killing capabilities of these immune cells and boosts antitumor immunity [32–34]. Additionally, the inhibition of SHP-1 can improve the ability of dendritic cells (DCs) to process and present tumor antigens, thereby enhancing antitumor activity [35].

## Osteosarcoma single-cell annotation and the key regulatory role of the PTPN family subgroups

We initially conducted dimensionality reduction and clustering on osteosarcoma single-cell data, followed by precise annotation based on marker genes specific to particular cell populations [36]. Using a t-SNE plot (Fig 3A), we visualized primary cell types in osteosarcoma single-cell samples, which included NK/T cells, M2-type tumor-associated macrophages, osteoclasts, cancer-associated fibroblasts, M1-type tumor-associated macrophages, monocytes, osteoblasts, B cells, endothelial cells, and plasma cells. A heatmap (Fig 3B) provided detailed expression profiles of marker genes for each cell type. To investigate the role of the PTPN family in immune regulation, we applied AUCell for categorical analysis of the single-cell data. With an AUC threshold of 0.022, 13,402 cells were identified as Active (Fig 3C). A subsequent t-SNE plot (Fig 3D) illustrated the distribution of the PTPN family between Active and Inactive groups. A bar chart (Fig 3E) indicated that the active population was predominantly composed of T cells, monocytes, M1-type tumor-associated macrophages (M1_TAM), and M2-type tumor-associated macrophages (M2_TAM). Notably, significant changes in population proportions for T cells, monocytes, M1_TAM, and M2_TAM were observed between Active and Inactive groups. Differential gene expression analysis revealed significant differences in genes such as PTPN1, PTPN2, PTPN6, PTPN7, PTPN12, and PTPN22 between the two groups. Particularly, PTPN6 was highly expressed in T cells, monocytes, M1_TAM, and

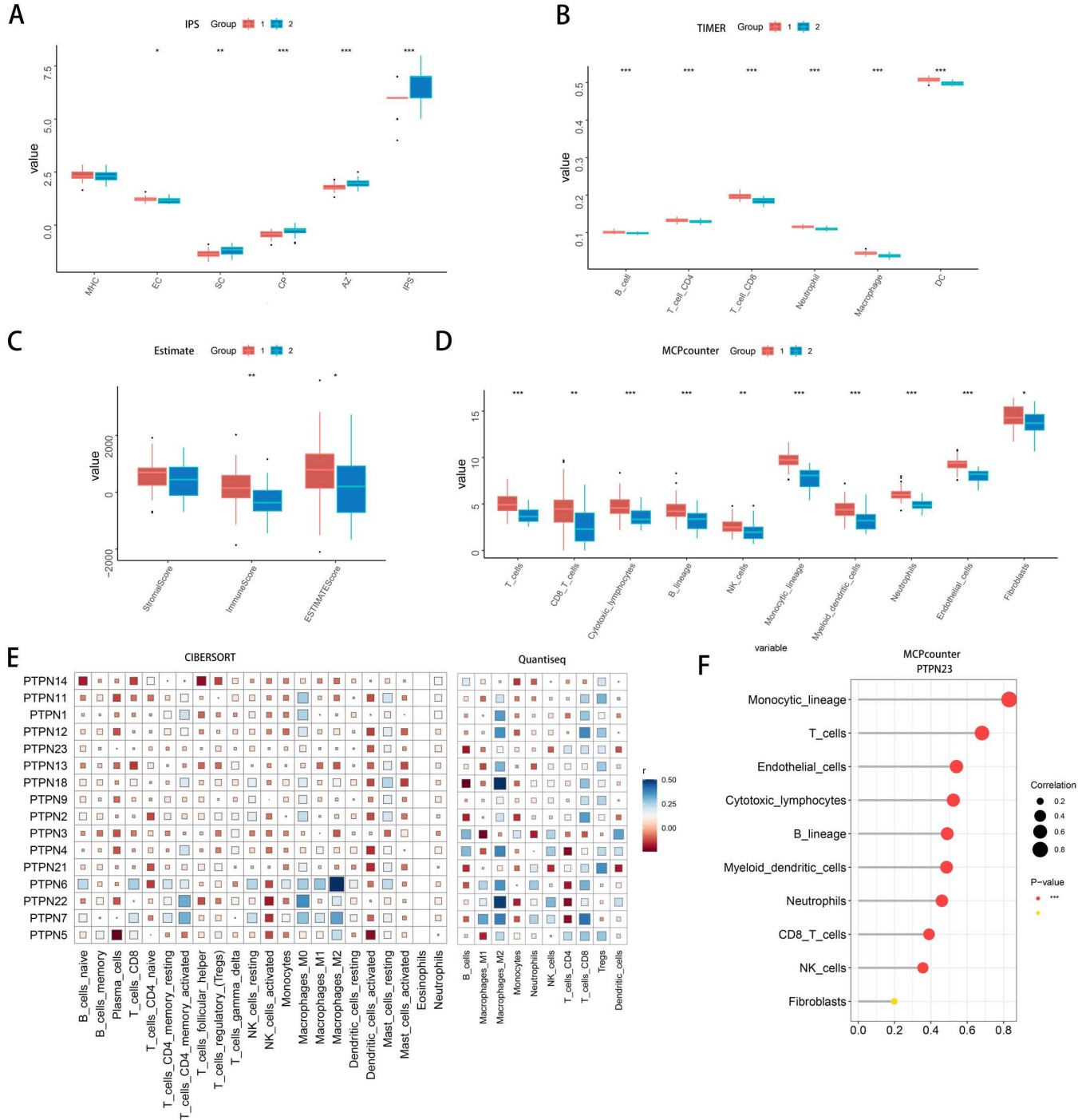

**Fig 2. Analysis of Immune Infiltration and PTPN Gene Expression in Clusters 1 and 2.** (A-D) Boxplots show differential expression of immune markers and enrichment scores between Cluster 1 (red) and Cluster 2 (blue). *Significance:* *p < 0.05, **p < 0.01, ***p < 0.001. (E) Correlation heatmap of immune infiltration types (CIBERSORT and Quantiseq) vs. PTPN family genes. (F) Enrichment scatter plot of immune cell types vs. enrichment levels.

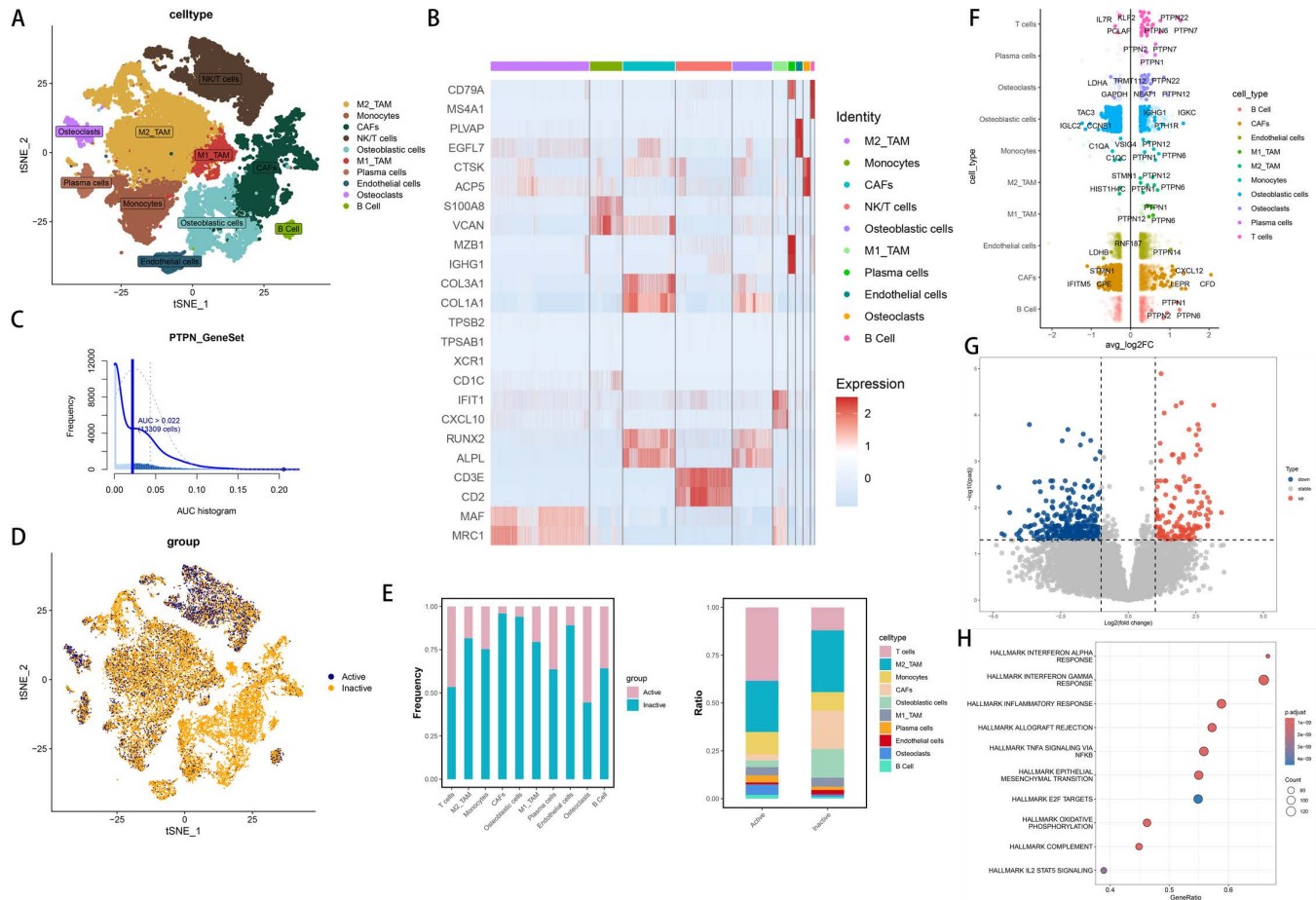

**Fig 3. Analysis of Cellular Heterogeneity and Differential Gene Expression in Active and Inactive Groups.** (A) tSNE plot show 10 cellular clusters by cell type. (B) Heatmap of marker gene expression across clusters. (C) The bar chart illustrates that at an AUC threshold of 0.022, 13,402 cells were identified as Active using AUCell. (D) tSNE plot differentiating Active (blue) and Inactive (yellow) groups. (E) Stacked bar chart of cellular subset proportions in Active and Inactive groups. (F) The volcano plots of different cell types illustrate the differentially expressed genes (DEGs) between the Active and Inactive groups. (G) Volcano plots of differentially expressed genes (DEGs) between Active and Inactive groups were generated based on pseudobulks. (H) Dot plots of Hallmark gene set enrichment analysis.

M2_TAM (Fig 3F), suggesting its crucial role in regulating the function and behavior of these immune cells. Additionally, we conducted a pseudobulks analysis on the single-cell data and performed differential analysis using DESeq2 with criteria set at (|logFC| > 1, p.value < 0.05). The results were presented in a volcano plot (Fig 3G). To further investigate the Active and Inactive states, we performed enrichment analysis using the Hallmark dataset, revealing correlations with the inflammatory regulation pathway, interferon response, TNF alpha signaling, and the EMT pathway(Fig 3H).

## Clustering analysis of t cells in Active and Inactive states

To further investigate the regulatory role of T cells in the Active and Inactive states, we conducted a detailed clustering analysis of T cells. We utilized specific markers for CD4[+] T cells (CD4, IL2RA, FOXP3) and CD8[+] T cells (CD8A, CD8B, GZMB) to facilitate this analysis (S2C Fig). The clustering results clearly identified the distribution of CD4[+] T cells and CD8[+] T cells (S2A and S2B Fig). Furthermore, we employed bar charts to illustrate the differences in the proportions of CD4[+] T cells and CD8[+] T cells between the Active and Inactive states. The results showed that the proportion of CD8[+] T

cells significantly increased in the Active state, while the proportion of CD4+ T cells was higher in the Inactive state (S2D and S2E Fig).

## CD4+ T cell subsets in osteosarcoma and immune regulatory mechanisms

We mapped CD4+ cells using projectSVR, subdividing them into three types: Tn (naive), Tm (memory), and TNFRSF9+ Treg (Fig 4A). Differential analysis revealed that Tn repressed the TNFRSF gene set, Tm activated JUND and KLRB1, while TNFRSF9+ Treg activated the TNFRSF gene set (S3A Fig). Enrichment analysis showed that TNFRSF9+ Tregs were enriched in pathways related to immune activation, including "Activation of immune response signaling," "Immune response activated by cell surface receptor," and "Canonical NF-kappaB signaling." (S4A Fig). TNFRSF9 (4−1BB) is a member of the TNF receptor family [37] and is recognized for its role in regulating T cell activation and function by activating key signaling pathways, such as NF-kappaB [38,39]. Our findings suggest that TNFRSF9+ Tregs contribute to immune suppression and may enhance their survival in the tumor microenvironment through activated signaling pathways, potentially modulating immune responses. In the active state, both Tm and TNFRSF9+ Tregs significantly increased, while Tn predominated in the inactive state (Fig 4B). Using Monocle2 for pseudotime analysis, we found that Tn differentiates into Tm and TNFRSF9+ Tregs over time (Fig 4C). CytoTRACE analysis supported this finding, indicating that Tn has a higher differentiation potential compared to Tm and TNFRSF9+ Tregs (S1B Fig). Pseudotime analysis of the PTPN family revealed that PTPN1, PTPN6, and PTPN22 are highly expressed in TNFRSF9+ Tregs, whereas PTPN2, PTPN4, and PTPN7 exhibited decreased expression in this subset (Fig 4D). Hallmark pathway enrichment analysis of TNFRSF9+ Tregs in different states indicated the activation of pathways like TNFA_SIGNALING_VIA_NFKB and OXIDATIVE_PHOSPHORYLATION (Fig 4E). Transcription factor analysis highlighted associations with ELF1, NFKB1, NFKB2, IRF8, and FOSL2 (Fig 4F), suggesting that the regulatory role of TNFRSF9+ Tregs is linked to inflammatory pathways and specific transcription factors. To further analyze the signaling interactions between CD4+ T cells in Active and Inactive states, we compared the interaction counts between the two groups(S5A Fig). In the signaling regulation induced by TNFRSF9+ Tregs, the signals of SELPLG-SELL and HLA-DR-CD4 were significantly enhanced, which may facilitate T cell adhesion and migration within the vasculature, thereby boosting T cell activation and immune response. Conversely, the interaction of HLA-DR-CD8 was diminished, which could suppress the functionality of CD8+ T cells, subsequently impacting their cytotoxicity against tumor cells(S5B Fig). In the signaling regulation induced by Tn, we found that it primarily activated the interaction between CLEC2B and KLRB1, while the signaling of HLA-DR-CD4 was inhibited(S5C Fig). Similarly, in the signaling regulation induced by Tm, CLEC2B-KLRB1 was also involved in the activation of self-regulatory processes(S5D Fig).

## Differentiation and functional characteristics of CD8+ T cell subsets in osteosarcoma immune regulation

We performed reference mapping on CD8+ T cells using ProjectSVR, identifying five subtypes: GZMK+ early Tem, Temra, Tex, ISG+ CD8+ T cells, and GZMK+ Tem (Fig 5A). Differential analysis validated the clustering of these subsets (S3B Fig), while enrichment analysis revealed that ISG+ CD8+ T cells were associated with viral response pathways, suggesting a role in immune surveillance. Temra cells were found to be enriched in pathways related to immune functions, indicating strong cytotoxic capabilities and the potential for antitumor immunity alongside NK cells (S4B Fig). In the Active state, we observed increased proportions of Temra, Tex, and ISG+ CD8+ T cells, with GZMK+ early Tem predominating (Fig 5B). Pseudotime analysis indicated that GZMK+ early Tem is an early differentiation stage, evolving into Temra, Tex, ISG+ CD8+ T cells, and GZMK+ Tem over time (Fig 5C). CytoTRACE supported this observation, indicating a higher differentiation potential in GZMK+ early Tem and GZMK+ Tem (S1A Fig). Analysis of the PTPN family revealed that PTPN6 expression increased with differentiation, while PTPN1, PTPN2, PTPN4, PTPN7, and PTPN22 were highly expressed in undifferentiated CD8+ T cells and decreased in ISG+ CD8+ T cells (Fig 5D). This suggests that the PTPN family regulates CD8+ T cell function and differentiation, thereby impacting immune responses in the tumor microenvironment. In the Active

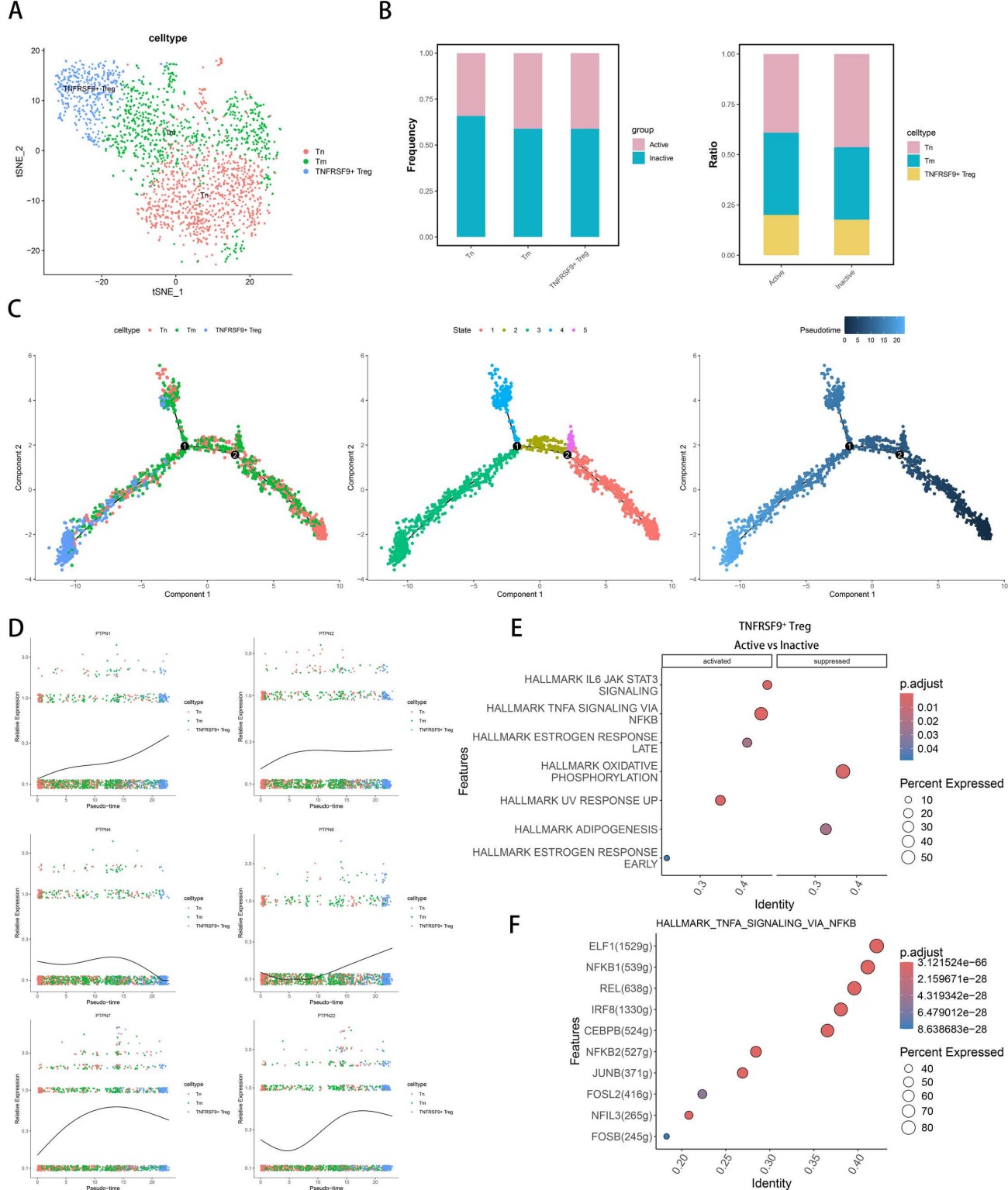

**Fig 4. Analysis of CD4⁺ T Cell Subpopulations and Functional States in Active and Inactive Groups.** (A) tSNE plot of CD4⁺ T cell subsets: Tn, Tm, and Treg. (B) Stacked bar charts showing CD4⁺ T cell subset proportions between Active and Inactive groups. (C) Scatter plot of CD4⁺ T cell trajectories from Monocle2 by subtype and pseudotime. (D) Scatter plot of PTPN gene expression across CD4⁺ T cell subsets along pseudotime. (E) Enrichment

scatter plot of pathway changes in TNFRSF9+ Treg cells between groups. (F) Enrichment scatter plot of transcriptional regulators in the TNFA_signaling_via_NFkB pathway for Treg cells.

state, Temra exhibited inhibited apoptosis (Fig 5E), with transcription factor analysis identifying SPI1, ELF1, and JUND as key regulators (Fig 5F). To further analyze the signaling interactions between CD8+ T cells in Active and Inactive states, we compared the interaction counts between the two groups and found that there were numerous signaling exchanges between them. In the signaling regulation induced by GZMK+ early Tem, significant activation was observed in the HLA-A to HLA-C CD8A signaling pathways, affecting Tex and ISG+ CD8+ T cells. Notably, similar regulatory effects were observed in Temra, Tex, and ISG+ CD8+ T cells(S5G-K Fig). These results indicate that the signaling interactions among different subsets of CD8+ T cells exhibit comparable patterns across different states, suggesting their potential role in modulating the tumor immune microenvironment.

## Mononuclear macrophage subsets and regulatory roles in the tumor microenvironment

We subdivided mononuclear macrophages into three subsets: monocytes, M1-type tumor-associated macrophages (M1_TAM), and M2-type tumor-associated macrophages (M2_TAM) using dimensionality reduction and clustering analysis (Fig 6A). Differential analysis confirmed the clustering of these subsets (S3C Fig), while enrichment analysis revealed that M2_TAM cells are linked to pathways that promote tumor growth, including cell cycle regulation and DNA damage response (S4C Fig). In the active state, the proportions of monocyte increased significantly, while those of M2_TAM decreased (Fig 6B). Pseudotime analysis indicated that monocytes are undifferentiated and gradually differentiate into M1_TAM and M2_TAM over time (Fig 6C). CytoTRACE supported this observation, differentiation a higher differentiation potential for monocytes compared to M1_TAM and M2_TAM (S1C Fig). Analysis of the PTPN family revealed significant regulatory roles for PTPN2, PTPN6, PTPN9, PTPN11, PTPN16, and PTPN22 across these subsets (Fig 6D), with PTPN6 being more highly expressed in M2_TAM. Pathway enrichment between Active and Inactive states showed that the E2F TARGETS and G2M CHECKPOINT pathways in M2_TAM were inhibited, suggesting a suppression of their proliferative function during the active state (Fig 6E). We hypothesize that the expression of PTPN family may inhibit tumor progression by enhancing immune responses. Furthermore, transcription factor prediction for E2F TARGETS identified E2F1, E2F2, E2F8, and MYBL2 as potential regulators of these pathways (Fig 6F). To further analyze the signaling interactions between mononuclear macrophages in active and inactive states, we compared the interaction counts between the two groups and identified numerous signaling exchanges. In monocytes, upregulation is primarily mediated by the SPP1-CD44 interaction, which plays a regulatory role. Conversely, downregulation mainly involves the HLA-DRA-CD4 interaction with both M1 and M2 tumor-associated macrophages (TAMs) (S5L Fig). In M1 TAMs, upregulation is also primarily driven by the SPP1-CD44 interaction with monocytes, while downregulation occurs through HLA-DRA-CD4 interactions with both M1 and M2 TAMs (S5M Fig). Similarly, in M2 TAMs, upregulation is again predominantly mediated by SPP1-CD44 interactions with monocytes, whereas downregulation is primarily regulated through HLA-DRA-CD4 interactions with M1 and M2 TAMs (S5N Fig). Overall, these findings suggest that the SPP1-CD44 interaction plays a significant role in the upregulation of signaling among these cell types, while HLA-DRA-CD4 is crucial for downregulation, highlighting the complexity of signaling dynamics within the tumor microenvironment.

## Analysis of PTPN23 in pan-cancer

We conducted a pan-cancer analysis to examine the role of PTPN23 in various malignant tumors using the TCGA pan-cancer database. Our analysis revealed that PTPN23 is generally under-expressed across different tissues and cancer types (Fig 7A). We observed significant differential expression between cancerous and non-cancerous tissues, with higher PTPN23 levels noted in esophageal adenocarcinoma (ESCA), stomach adenocarcinoma (STAD), low-grade

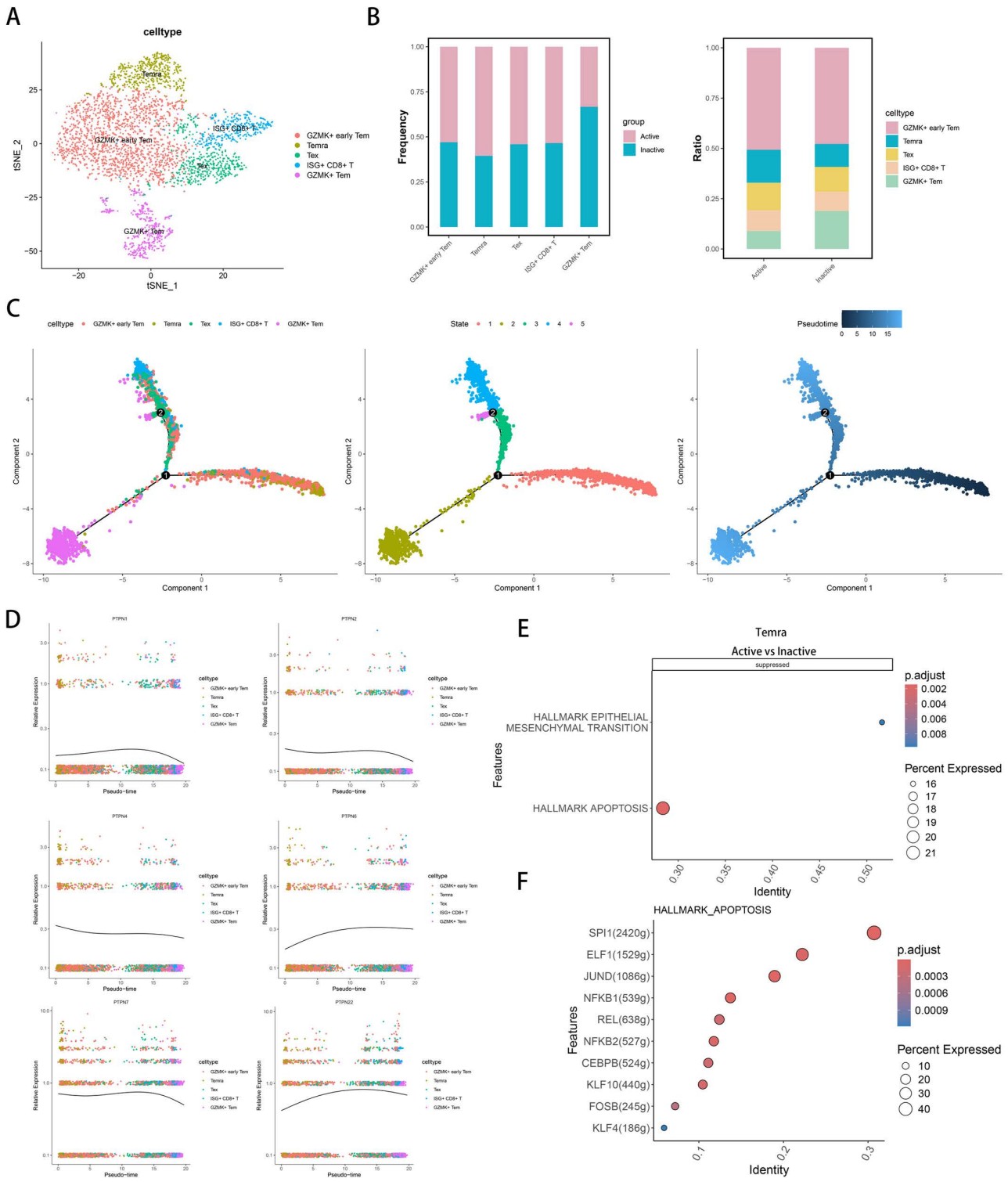

**Fig 5. Characterization and Functional Analysis of CD8+ T Cell Subsets in Active and Inactive Groups.** (A) tSNE plot of CD8+ T cell subsets. (B) Stacked bar charts of CD8+ T cell subset proportions between Active and Inactive groups. (C) Scatter plot of CD8+ T cell pseudotime trajectory from Monocle2. (D) Scatter plot of PTPN gene expression dynamics across CD8+ T cells along pseudotime. (E) Enrichment scatter plot of pathway changes in Temra cells. (F) Enrichment scatter plot of transcriptional regulators in the Apoptosis pathway for Temra cells.

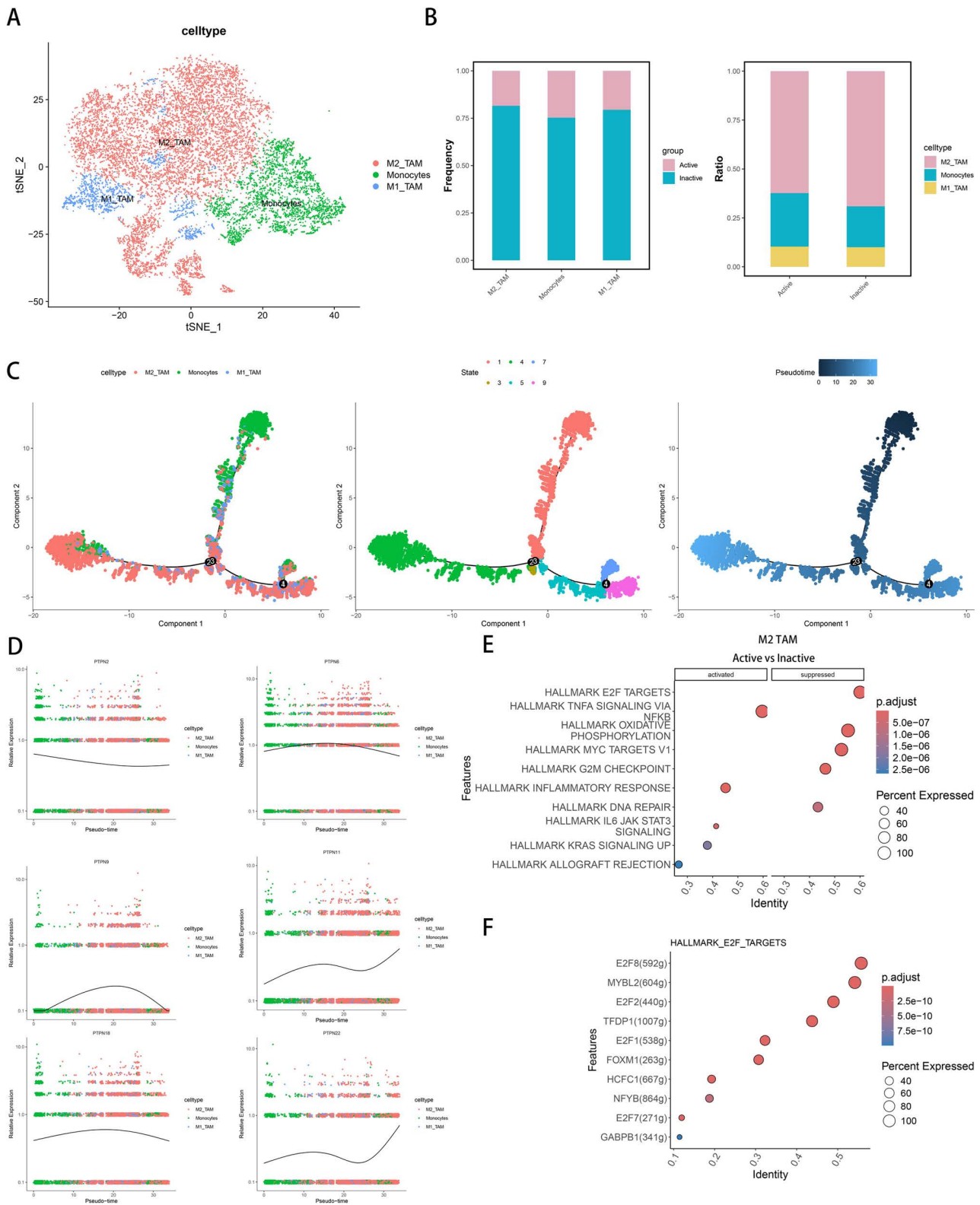

**Fig 6. Heterogeneity and Functional Trajectory of Mononuclear Phagocytes in Active and Inactive Groups.** (A) tSNE plot of mononuclear plago-cyte subsets: monocytes,M1_TAM, and M2_TAM. (B) Stacked bar charts showing proportional changes of mononuclear phagocyte subsets between

Active and Inactive groups. (C) Scatter plot of pseudotemporal trajectories of mononuclear phagocyte subsets from Monocle2. (D) Scatter plot of PTPN gene expression dynamics across mononuclear phagocyte subsets along the pseudotemporal trajectory. (E) Enrichment scatter plot of pathway changes in M2_TAM between Active and Inactive groups. (F) Enrichment scatter plot of transcriptional regulators in the E2F pathway for M2_TAM.

glioma (LGG), and colon adenocarcinoma (COAD), while other cancers displayed low expression levels(Fig 7B). To validate these findings, we combined GTEx and TCGA datasets, which affirming significant differential expression of PTPN23 across multiple cancer types (Fig 7C). Additionally, we performed univariable analysis on 33 cancer types and found that PTPN23 significantly impacts overall survival (OS) in several cancers, including kidney chromophobe (KICH), sarcoma (SARC), skin cutaneous melanoma (SKCM), thyroid carcinoma (THCA), and uveal melanoma (UVM) (Fig. 7F). Hallmark gene set enrichment analysis indicated that PTPN23 influences important pathways, including the interferon γ response, inflammatory response, and IL2_STAT5 (Fig 7E). Furthermore, we identified a close association between PTPN23 and T cells, as well as monocytes/macrophages (Fig 7D).

### In Vitro validation of PTPN23

To investigate the effect of si-PTPN23 on the proliferation of the osteosarcoma cell lines 143B and SJSA-1, we performed CCK8 assays at 24, 48, and 72 hours. The results showed that si-PTPN23 significantly promoted cell proliferation (Fig 8A). Plate cloning experiments confirmed that si-PTPN23 enhanced cloning efficiency compared to si-NC group (Fig 8B), while EDU assays indicated an increased in proliferative activity (Fig 8C). The knockdown efficiency of PTPN23 was approximately 30%, confirming effective transfection. To further validate that this enhanced proliferation was associated with cell cycle regulation, we conducted cell cycle assays. These assays revealed an increase in the proportion of cells in the S phase following si-PTPN23 treatment compared to si-NC(S6A to S6D Fig), corroborating the role of si-PTPN23 in promoting cell proliferation. Western blot analysis showed significant upregulation of the proliferation marker PCNA following si-PTPN23 transfection. Additionally, we explored the association with the JAK-IL6-STAT3 signaling pathway and found increased levels of IL6, STAT3, and phosphorylated STAT3 (p-STAT3) in Western blot experiments (Fig 8D). These findings suggests that the PTPN family may regulate the tumor microenvironment through the JAK-IL6-STAT3 pathway, thereby promoting osteosarcoma cell proliferation.

### Discussion

Osteosarcoma is a malignant bone tumor that predominantly affects adolescents and young adults. This cancer is characterized by rapid growth and a high potential for metastasis. The PTPN family, known for regulating oncogenic pathways through the dephosphorylation of proteins, plays a critical role in tumor progression and the development of treatment resistance. Targeting specific PTPN members has the potential to enhance antitumor immunity, positioning them as promising therapeutic targets.

In this study, we identified two distinct molecular subtypes of osteosarcoma, referred to as Cluster 1 and Cluster 2, which display varying survival rates. This classification was achieved using ConsensusClusterPlus and AUcell analyses on both bulk and single-cell RNA sequencing data. These findings underscore the significant role of PTPNs in the development and prognosis of osteosarcoma, suggesting that further exploration of these molecules could lead to more effective therapeutic strategies. Notably, Cluster 2 demonstrates immunosuppressive characteristics associated with the enrichment of the interferon signaling pathway and the regulation of immune cells. Despite this immunosuppression, a subset of CD4+ and CD8+ T cells exhibit signs of activation, indicating a potential for effective antitumor responses. This suggests that, although the tumor microenvironment in Cluster 2 may be predominantly immunosuppressive, there remains a possibility for harnessing the activity of these activated T cells to mount a robust immune response against the osteosarcoma.

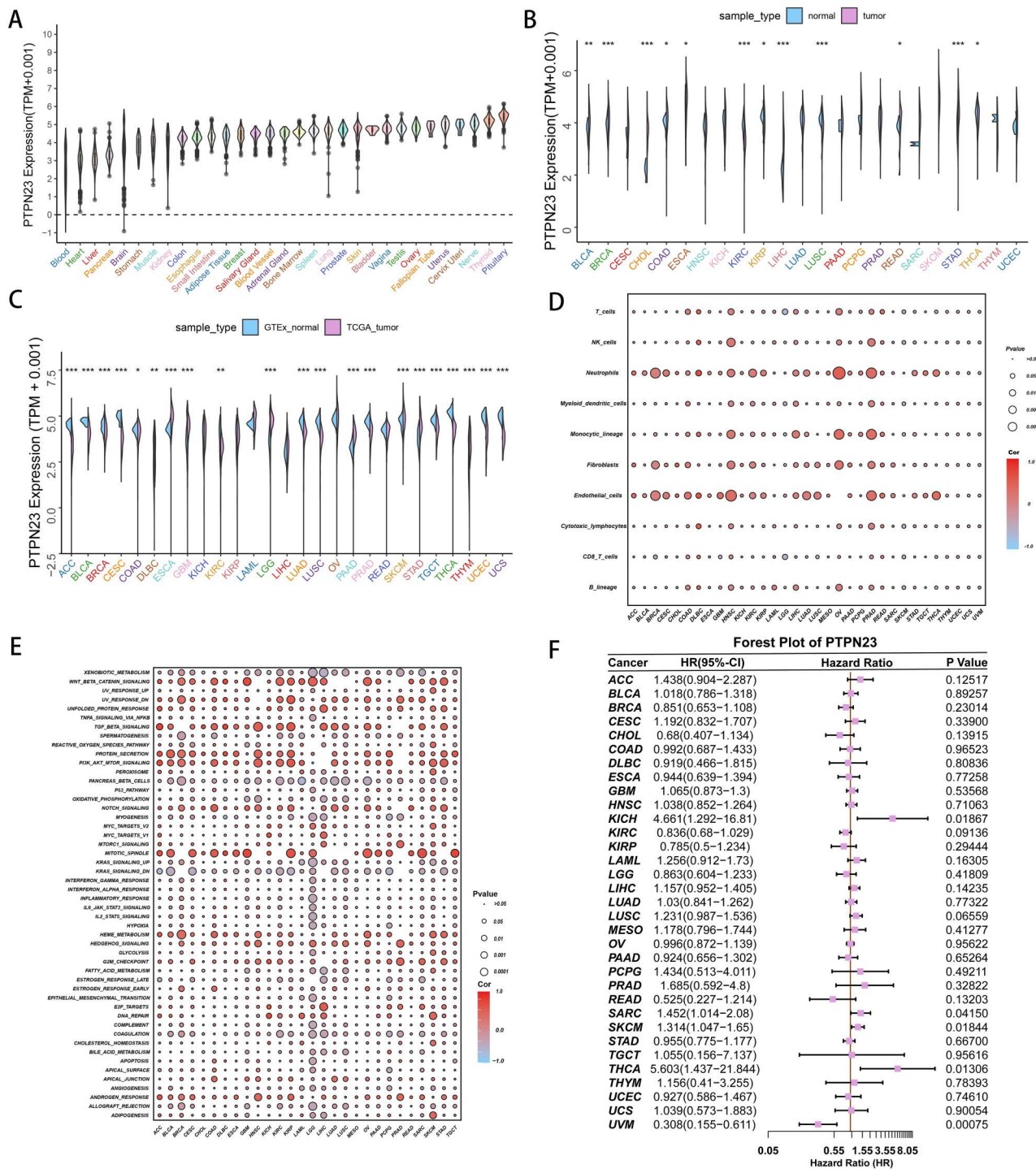

**Fig 7. Pan-Cancer Analysis of PTPN23 Expression and Its Associations with Clinical Outcomes and Immune Infiltration.** (A) PTPN23 mRNA expression across cancer types in TCGA. (B) PTPN23 expression comparison between normal and tumor samples in TCGA. (C) PTPN23 expression in normal (GTEx) vs. tumor (TCGA) samples. (D) Correlation of PTPN23 expression with immune cell infiltration across cancers. (E) Correlation of PTPN23 expression with Hallmark gene sets and immune pathways. (F) Forest plot of PTPN23 prognostic significance across different cancers. *p < 0.05, **p < 0.01, ***p < 0.001, ****p < 0.0001.

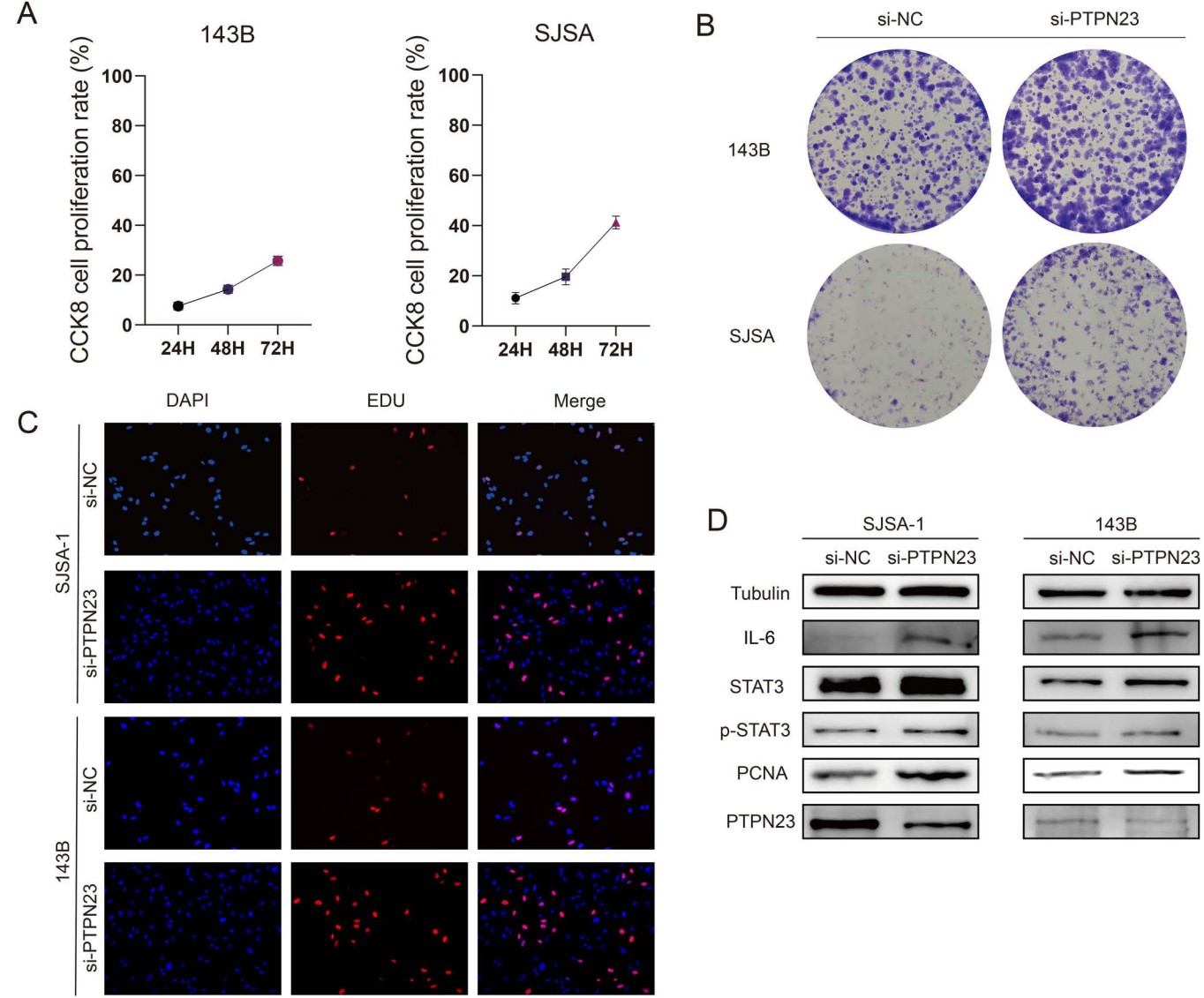

**Fig 8. Regulation of osteosarcoma cell proliferation by PTPN23.** (A) Absorbance at 450 nm after CCK8 treatment in PTPN23 groups; higher absorbance indicates more cell proliferation. (B) Plate cloning results for SRSF7 treatment groups. (C) Hoechst & EDU staining showing proliferative activity in PTPN23 groups was captured under a 20x (200μm) magnification. (D) Protein expression levels of PTPN23, PCNA, IL-6, STAT3, and p-STAT3 in SJSA-1 and 143B cells from knockdown and control groups. The raw data of the protein bands are stored in S8 Raw images.

To further investigate the regulatory role of the PTPN family on immune cells, we employed AUcell for enrichment scoring on single-cell osteosarcoma datasets, classifying the cells into Active and Inactive groups. Our analysis revealed significant regulatory effects among CD4+ T cells, CD8+ T cells, and mononuclear macrophages. Specifically, the PTPN family appears to inhibit CD4+ T cell activity, which contributes to immune evasion by the tumor. Conversely, the activation of CD8+ T cells, along with a reduction in M2-type tumor-associated macrophages (TAMs), is associated with enhanced tumor cell death.

Our analysis revealed that Cluster 2, which is associated with poor prognosis, displayed notable immunosuppressive traits within its immune cell populations. We specifically investigated three subsets of CD4+ T cells: naïve (Tn), memory

(Tm), and TNFRSF9⁺ regulatory T cells (Tregs). We observed a significant increase in the active memory T cell (Tm) and TNFRSF9⁺ Treg populations, highlighting their crucial roles in the immune response. Pseudotime analysis indicated a differentiation pathway progressing from Tn to Tm and TNFRSF9⁺ Tregs, further emphasizing the regulatory involvement of the PTPN family in this process. The elevated expression of PTPN family members within TNFRSF9⁺ Tregs suggests their substantial influence on the behavior of these immunosuppressive cells within the tumor microenvironment [40]. TNFRSF9⁺ Tregs, characterized by high expression of TNFRSF9 (4−1BB), are specialized in maintaining immune suppression [37]. Agonistic antibodies targeting 4−1BB have garnered considerable interest in immunotherapy research due to their effectiveness in inhibiting tumor growth in preclinical models [41]. However, the translation of 4−1BB immunotherapy into clinical settings poses challenges, particularly regarding the risk of uncontrolled toxicity [42,43]. Furthermore, the presence of TNFRSF9⁺ Tregs correlates with the composition of tumor-infiltrating lymphocytes and their ability to recognize antigens, marking them as valuable biomarkers for predicting responses to immune checkpoint inhibitors in melanoma patients [44]. Moreover, mechanisms of immune escape may exacerbate antitumor immune response suppression by increasing the proportion and activity of TNFRSF9⁺ Tregs [45]. Hallmark pathway enrichment analysis also revealed that the transition of TNFRSF9⁺ Tregs from inactive to active states is linked to inflammation-related pathways, such as TNF-alpha signaling via NF-kappaB and oxidative phosphorylation.

We identified five CD8⁺ T cell subtypes, with GZMK⁺ early effector memory T cells (Tem) playing a crucial role in differentiation. In the Active states, we observed an increase in TEMRA (T effector memory re-expressing CD45RA), Tex (exhausted T cells), and ISG⁺ T cells, which indicate key effector functions against tumors. Hallmark pathway analysis linked the rising levels of TEMRA cells to apoptosis inhibition, suggesting that they enhance tumor resistance and potentially suppress mechanisms of immune escape [46]. CD8⁺ TEMRA cells, which represent late-stage effector memory T cells characterized by high levels of perforin, are particularly noteworthy [47]. In gastric cancer patients, the upregulation of CD8⁺ TEMRA cells may serve as an indicator of active immune surveillance in the tumor microenvironment [48]. Furthermore, an increase in CD8⁺ TEMRA cells has been observed in metastatic lymph nodes (TDLN), correlating with shifts in other immune cell types, such as CD4⁺ T cells and regulatory T cells (Tregs). This alteration in immune composition is also associated with changes in the expression of immune checkpoint molecules like PD-1 and CTLA-4 [49].

Through single-cell transcriptome analysis, we identified three distinct macrophage subsets: monocytes, M1 tumor-associated macrophages (TAMs), and M2 TAMs. We observed an increase in monocyte proportions in the active immune state, accompanied by a decrease in M2 TAMs. Pseudotime analysis indicated a differentiation trajectory where monocytes evolve into M1 and M2 TAMs, highlighting their higher differentiation potential [50]. M2 TAMs, in particular, have been associated with clinical prognosis in osteosarcoma, often promoting an immunosuppressive microenvironment that facilitates cancer cell evasion of the immune system [51]. For instance, M2 macrophages that secrete IL-10 can downregulate IL-12 expression in dendritic cells, subsequently impairing the activation of CD8⁺ T cells, and thereby allowing tumor cells to evade immune detection [52,53]. However, our findings showed that despite the reduction in M2 macrophages, the antitumor activity of CD8⁺ TEMRA cells increased. This suggests that the decrease in M2 macrophages may have created a more favorable microenvironment for the expansion of CD8⁺ TEMRA cells, promoting a more effective antitumor immune response.

In our investigation of pseudotime gene expression patterns within the PTPN family, we found that elevated expression levels were predominantly present in TNFRSF9⁺ regulatory T cells (Tregs), CD8⁺ TEMRA cells, and M2 tumor-associated macrophages (TAMs). Notably, the expression of PTPN family members increased progressively along the pseudotime trajectory, indicating a potential role in the differentiation and functional regulation of these immune cell subsets. Furthermore, through cell communication analysis, we discovered significant receptor-ligand interactions between the Active and Inactive states of PTPN, indicating potential pathways for communication that may influence the immune landscape.

PTPN23 (Protein Tyrosine Phosphatase Non-Receptor Type 23) is a member of the protein tyrosine phosphatase family that exhibits a complex and multifaceted regulatory role in tumorigenesis. Its functions can act as both a tumor

suppressor and, through specific signaling pathways, indirectly influence tumor progression. In breast cancer, for instance, low expression of PTPN23 is significantly correlated with poor prognosis in patients [54]. Moreover, PTPN23 knockout mice display accelerated tumor growth and lung metastasis in transplantation models. Additionally, PTPN23 expression has been positively associated with PD-L1, suggesting that it may facilitate immune evasion by modulating T cell activity. In colorectal cancer (CRC), the loss of PTPN23 leads to decreased expression of E-cadherin, along with upregulation of epithelial-mesenchymal transition (EMT) markers such as Vimentin and SNAIL, promoting cell migration [55]. Currently, the use of SND1 inhibitors (such as pdTp) or miR-142-3p antagonists has been identified as a strategy to upregulate PTPN23, thus inhibiting tumor growth [56]. Furthermore, in EGFR-mutant lung cancer, combining WDR4 inhibitors with EGFR TKIs has shown potential in overcoming resistance [57].

In our bulk study, we identified that PTPN23 exhibited strong model performance in random forest analysis, revealing significant enrichment of the IL6-STAT3 pathway within the PTPN family. Our pan-cancer analysis indicated that low PTPN23 expression correlates with poor prognosis across various tumors, suggesting it role as a tumor suppressor, particularly in KICH, SARC, SKCM, THCA, and UVM. Enrichment analysis linked PTPN23 to interferon signaling and inflammation, emphasizing its involvement in the tumor immune microenvironment. Additionally, silencing PTPN23 promotes osteosarcoma cell proliferation by modulating cell cycle proteins, such as PCNA, indicating its mechanistic role in tumorigenesis and its potential as a therapeutic target. We also identified associations with pathways like IL-6/JAK/STAT3 through differential enrichment analysis of Cluster 1 and Cluster 2.

In future studies, we aim to conduct small animal experiments to investigate the effects of PTPN23 inhibition and overexpression on tumor growth, metastasis, and immune responses. We will employ techniques such as small animal imaging and histological analysis to evaluate the progression of tumors in response to these manipulations. Furthermore, we plan to integrate single-cell RNA sequencing technology to comprehensively analyze the changes in the tumor microenvironment following both PTPN23 inhibition and overexpression. This multifaceted approach will help us understand the role of PTPN23 in tumor dynamics and immune interactions in greater detail. Our research suggests that the PTPN family, particularly PTPN23, plays a crucial role in tumor immune evasion. Differential expression of PTPN23 significantly influences the mechanisms by which tumors escape immune surveillance. Inhibiting PTPN23 may lead to the dysregulation of downstream signaling pathways, potentially increasing the proliferation and survival of tumor cells while simultaneously weakening immune effector responses. This dysregulation could enable tumors to avoid detection by the immune system more effectively. Conversely, overexpressing PTPN23 might counteract these effects, potentially restoring immune responsiveness. This intricate balance underscores the importance of PTPN23 as a therapeutic target and its potential implications for enhancing anti-tumor immunity.

## Conclusions

This study explores the role of the PTPN family in the single-cell composition of osteosarcoma and reveals that differences in cell population proportions indicate the regulatory potential of the immune microenvironment, primarily involving T cells and monocyte-derived macrophages. Through bioinformatics analysis, our results demonstrate that the activation of the PTPN family is associated with significant changes in the proportions of TNFRSF9$^+$ regulatory T cells (Tregs), effector memory CD8$^+$ T cells re-expressing CD45RA (TEMRA), and M2 tumor-associated macrophages (TAMs), which also exhibit distinct differentiation trends. According to previous studies, these cellular states play critical roles in regulating tumor cell survival within the osteosarcoma microenvironment.

In our comprehensive analysis, PTPN23 was identified as being linked to the regulation of tumor survival. To further investigate this finding, we conducted silencing experiments on PTPN23, which revealed its functional role in promoting tumor growth, potentially mediated through the IL-6/JAK/STAT3 signaling pathway. Looking ahead, it is essential to conduct further research on the PTPN gene family in animal models to deepen our understanding of their regulatory mechanisms within the osteosarcoma immune microenvironment, thereby enhancing our overall comprehension of tumor

biology. Consequently, improving our understanding of the tumor microenvironment will aid in identifying osteosarcoma within the immune landscape and provide a basis for developing more targeted therapeutic strategies.

## Supporting information

**S1 Fig. CytoTRACE illustrates changes in differentiation states of mononuclear macrophages, CD4⁺T cells, and CD8⁺T cells.** (A) UMAP of mononuclear macrophages and T cell phenotypes (GZMK⁺ early Tem, Temra, Tex, ISG+CD8⁺T, GZMK⁺ Tem) with CytoTRACE values indicating differentiation states. (B) UMAP of CD4⁺T cells, showing naive (Tn), memory (Tm), and TNFRSF9⁺Treg populations, along with corresponding CytoTRACE differentiation levels. (C) UMAP of mononuclear phagocytes (M2-TAM, monocytes, M1-TAM) with CytoTRACE values illustrating their differentiation states.
(TIF)

**S2 Fig. Annotations and proportional changes for CD4⁺T cells and CD8⁺T cells.** (A) UMAP visualization showing clustering of CD4⁺ and CD8⁺T cells based on RNA expression profiles, with distinct color coding for different cell populations. (B) Cell type annotation indicating CD4⁺T cells (in yellow) and CD8⁺T cells (in blue), highlighting their distribution within the UMAP. (C) Expression patterns of key markers (CD4, IL2RA, FOXP3, CD8A, GZMB) across the cell populations, with color intensity reflecting marker expression levels. (D) Bar graph illustrating the frequency of Active (red) and Inactive (blue) states for CD4⁺ and CD8⁺T cells. (E) Bar graph showing the ratio of Active and Inactive states for CD4⁺ and CD8⁺T cells, indicating overall proportions between the two cell types.
(TIF)

**S3 Fig. Volcano plots of differentially expressed genes for the cell populations of mononuclear macrophages, CD4⁺T cells, and CD8⁺T cells.** (A) Volcano plots showing differentially expressed genes for CD4⁺T cell subsets, including naive (Tn), memory (Tm), and TNFRSF9⁺Treg cells, with key genes highlighted. (B) Volcano plots for CD8⁺T cell populations, showcasing GZMK⁺ early Tem, Temra, Tex, ISG⁺CD8⁺, and GZMK⁺ Tem subsets, emphasizing significant gene expression changes. (C) Volcano plots representing mononuclear macrophage populations, including M2_TAM, monocytes, and M1_TAM, identifying key genes that define their transcriptional profiles.
(TIF)

**S4 Fig. Heatmap and enrichment analysis of cell populations: mononuclear macrophages, CD4⁺T cells, and CD8⁺T cells.** (A) Heatmap and enrichment analysis of gene expression for mononuclear macrophages, highlighting key biological processes, pathways, and enriched terms associated with their activity. (B) Heatmap and enrichment analysis for CD4⁺T cell subsets, including naive (Tn), memory (Tm), and TNFRSF9⁺Treg cells, showcasing significant gene expression and enriched pathways relevant to their functions. (C) Heatmap and enrichment analysis of CD8⁺T cell populations, featuring GZMK⁺ early Tem, Temra, Tex, ISG⁺CD8⁺, and GZMK⁺ Tem subsets, illustrating differences in gene expression and pathway enrichment critical for their immune roles.
(TIF)

**S5 Fig. Monocytes、CD4⁺T cells and CD8⁺T cells exhibited distinct signal communication patterns in both Active and Inactive states.** (A-C) the interactions among different cell types under these two conditions. (D-F) Scatter plots showing the upregulated and downregulated ligands and receptors for naive T cells, memory T cells, and regulatory T cells. Additionally, panels (G-K) Scatter plots that depict the upregulated and downregulated ligands and receptors for GZMK⁺ early memory T cells, TEMRA, GZMK⁺ memory T cells, exhausted T cells, and ISG⁺CD8⁺T cells. (L-N) The upregulated and downregulated ligands and receptors for monocytes, M1 tumor-associated macrophages (TAMs), and M2 TAMs.
(TIF)

**S6 Fig. Impact of si-PTPN23 on Cell Cycle Distribution in Osteosarcoma Cell Lines SJSA-1 and 143B.** (A-B) Flow cytometric analysis of the cell cycle distribution in SJSA-1 and 143B cells. The left panel represents the negative control (NC), while the right panel shows the results for si-PTPN23 treatment. The different phases of the cell cycle are indicated by the various colors: G0/G1 (blue), S (green), and G2/M (orange). (C-D) Bar graph comparing the cell cycle phase distribution in SJSA-1 and 143B cells between the NC and si-PTPN23 groups. The percentage of cells in G0/G1, S, and G2/M phases is shown, with asterisks indicating statistical significance ($*p < 0.05$).
(TIF)

**S7 file. t-SNE embedding illustrates the distribution of cell populations.**
(ZIP)

**S8 Raw images.**
(PDF)

## Author contributions

**Conceptualization:** Changhai Long, Biao Ma.

**Data curation:** Changhai Long, Biao Ma.

**Formal analysis:** Changhai Long, Biao Ma, Xingshun Zhong.

**Investigation:** Changhai Long, Biao Ma, Kai Li.

**Methodology:** Changhai Long, Biao Ma.

**Project administration:** Changhai Long, Biao Ma, Sijing Liu.

**Resources:** Biao Ma.

**Software:** Changhai Long, Biao Ma, Mingzhi Zou.

**Supervision:** Changhai Long, Biao Ma.

**Writing – original draft:** Changhai Long, Biao Ma.

**Writing – review & editing:** Changhai Long, Biao Ma.

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
