## [Decision Letter · Decision Letter 0]

Dear Dr. Liu,

Thank you for submitting your manuscript to PLOS ONE. After careful consideration, we feel that it has merit but does not fully meet PLOS ONE’s publication criteria as it currently stands. Therefore, we invite you to submit a revised version of the manuscript that addresses the points raised during the review process.

We look forward to receiving your revised manuscript.

Kind regards,

Jinhui Liu

Academic Editor

PLOS ONE

Journal Requirements:

Additional Editor Comments:

Authors should revise according to the suggestions of reviewers. The modifications should be marked. A point to point response letter is needed.

Reviewers' comments:

Reviewer's Responses to Questions

**Comments to the Author**

1. Is the manuscript technically sound, and do the data support the conclusions?

Reviewer #1: Partly

2. Has the statistical analysis been performed appropriately and rigorously?

Reviewer #1: Yes

3. Have the authors made all data underlying the findings in their manuscript fully available?

Reviewer #1: Yes

4. Is the manuscript presented in an intelligible fashion and written in standard English?

Reviewer #1: No

Reviewer #1: The authors have carried out a series of studies on osteosarcoma using multi-omics analysis, which is quite interesting. There is few mistake in the manuscript and the experimental results show good and new results. So I recommend to you that this manuscript should receive revision. The following are the questions and some mistakes in this manuscript:

1.It is noted that your manuscript needs careful editing by someone with expertise in technical English editing paying particular attention to English grammar, spelling, and sentence structure so that the goals and results of the study are clear to the reader.

2.All the blots/gels must have the molecular weight/bp indicated.

3.Include scale bars to all microscopy images.

4.Small interfering RNA (siRNA) sequences need to be clearly listed.

Although the author conducted experiments for verification, I think the simple in vitro cell experiment is still not convincing enough, and I suggest adding in vivo experiments to further verify the relevant conclusions.

**Do you want your identity to be public for this peer review?** For information about this choice, including consent withdrawal, please see our Privacy Policy

Reviewer #1: No

---

## [Author Response · Author response to Decision Letter 1]

15 Jan 2025

Journal Requirements:

Answer�I have revised the manuscript in accordance with the requirements of your journal.

2.PLOS requires an ORCID iD for the corresponding author in Editorial Manager on papers submitted after December 6th, 2016. Please ensure that you have an ORCID iD and that it is validated in Editorial Manager. To do this, go to ‘Update my Information’ (in the upper left-hand corner of the main menu), and click on the Fetch/Validate link next to the ORCID field. This will take you to the ORCID site and allow you to create a new iD or authenticate a pre-existing iD in Editorial Manager.

Answer�I have added the ORCID iD for the corresponding author as per the journal's requirements. Thank you for your guidance.

Answer Thank you for your observation. I have revised the manuscript to ensure that the grant information in the ‘Funding Information’ and ‘Financial Disclosure’ sections matches correctly.

3.PLOS ONE now requires that authors provide the original uncropped and unadjusted images underlying all blot or gel results reported in a submission’s figures or Supporting Information files. This policy and the journal’s other requirements for blot/gel reporting and figure preparation are described in detail at https://journals.plos.org/plosone/s/figures#loc-blot-and-gel-reporting-requirements and https://journals.plos.org/plosone/s/figures#loc-preparing-figures-from-image-files. When you submit your revised manuscript, please ensure that your figures adhere fully to these guidelines and provide the original underlying images for all blot or gel data reported in your submission. See the following link for instructions on providing the original image data: https://journals.plos.org/plosone/s/figures#loc-original-images-for-blots-and-gels.   

Answer Thank you for your guidance regarding the submission requirements. I have added the original uncropped and unadjusted protein data to Supporting Information as S5 Fig. In my cover letter, I will indicate that the raw blot/gel image data is provided in the Supporting Information.

Answer Thank you for your suggestion. I have revised the manuscript to include captions for the Supporting Information files at the end and updated any in-text citations to ensure they match accordingly.

Response to Reviewers 1

Reviewer #1: The authors have carried out a series of studies on osteosarcoma using multi-omics analysis, which is quite interesting. There is few mistake in the manuscript and the experimental results show good and new results. So I recommend to you that this manuscript should receive revision. The following are the questions and some mistakes in this manuscript:

1.It is noted that your manuscript needs careful editing by someone with expertise in technical English editing paying particular attention to English grammar, spelling, and sentence structure so that the goals and results of the study are clear to the reader.

Answer Thank you for your feedback. I have carefully revised the manuscript to address issues related to grammar and sentence structure throughout the text, with specific changes highlighted in red.

2.All the blots/gels must have the molecular weight/bp indicated.

Answer We have added the molecular weight/bp information in Fig 8D.

3.Include scale bars to all microscopy images.

Answer�I have modified Fig 8C to include images captured at a scale of 200 µm.

4.Small interfering RNA (siRNA) sequences need to be clearly listed.

Answer Thank you for your valuable feedback. I have clearly listed the small interfering RNA (siRNA) sequences in the Methods section, specifically on lines 145-148.

Although the author conducted experiments for verification, I think the simple in vitro cell experiment is still not convincing enough, and I suggest adding in vivo experiments to further verify the relevant conclusions.

Answer Thank you for your insightful comment. I appreciate your suggestion regarding the inclusion of in vivo experiments to strengthen our findings. While we believe that the in vitro results provide important preliminary data, we acknowledge the value of in vivo validation. We will consider this recommendation for future work to enhance the robustness of our conclusions.

Journal Requirements 2:

1.Answer Thank you for your suggestion. I have revised the manuscript to include captions for the Supporting Information files at the end and updated any in-text citations to ensure they match accordingly. Additionally, I have included the S6 raw image.

2.Answer Thank you for your suggestion. I have added the raw data to the Supporting Information files in a ZIP format within the S6 file to comply with the minimal data set requirements.

---

## [Decision Letter · Decision Letter 1]

Dear Dr. Liu,

Thank you for submitting your manuscript to PLOS ONE. After careful consideration, we feel that it has merit but does not fully meet PLOS ONE’s publication criteria as it currently stands. Therefore, we invite you to submit a revised version of the manuscript that addresses the points raised during the review process.

We look forward to receiving your revised manuscript.

Kind regards,

Zhiwen Luo

Academic Editor

PLOS ONE

Journal Requirements:

Additional Editor Comments :

Thank you for submitting your manuscript to the Journal and as voucan see that the reviewer think your manuscript is interesting and provide valuable comments for your reference. Please submit the revised manuscript ASAP and also include a rebuttal that would clearly list all the responses to the reviewer's comments.

Reviewers' comments:

Reviewer's Responses to Questions

**Comments to the Author**

Reviewer #2: All comments have been addressed

2. Is the manuscript technically sound, and do the data support the conclusions?

Reviewer #2: Yes

3. Has the statistical analysis been performed appropriately and rigorously?

Reviewer #2: Yes

4. Have the authors made all data underlying the findings in their manuscript fully available?

Reviewer #2: Yes

5. Is the manuscript presented in an intelligible fashion and written in standard English?

Reviewer #2: Yes

Reviewer #2: We thank the authors for their submission, and recognize the hard work they put in preparing this manuscript for submission. It is a thorough study and seeks to leverage a large amount of data in order to identify possible therapeutic targets in osteosarcoma. There are, however, areas which need to be addressed to help the authors obtain their own goal within the current scope of this project to impact their field as they seek to do:

While attention has been paid to grammar and editing the English used in this manuscript, it would be worth having a native English speaker take a pass at ending some of the phrases used. Lines such as “The KEGG analysis” on line 196 and use of “based on your findings” on line 200 stand out as easily addressable grammar issues.

We appreciate the authors dual focus on T cells and macrophages and the role PTPNs may play in controlling osteosarcoma outcomes in their patient population. To that end, the data focuses on the differentiation from naïve T cells present in the samples into memory and regulatory T cells. However, the authors only use single biomarkers to identify these cell types and do not compare the subgroups across memory and regulatory T cells to see if there is an impact of the tumor microenvironment on their development. A fuller perspective on which T cell populations are affected by the tumor and its impact on cellular PTPN levels would be far informative for the reader. Similar questions may be asked of the monocyte-macrophage aspect of the study, as well.

The authors describe the use of gene enrichment to highlight PTPN family gene significance in active versus inactive gene signatures in Figure 3. However, what is lacking is a clear description of the active versus inactive signatures are, how they were validated in this disease type, and whether the genes identified in this study align with previous reports.

There is a distinct lack of control in the proliferation studies performed in this study, making it hard to derive validity and impact of the si-PTPN23 transfection into both osteosarcoma cell lines. Is the lack of proliferation due to cell death or simply paused by inhibition of PTPN23? The authors should consider using a non-dye based assay or measure cell viability at these time points to strength the data set.

Echoing the previous manuscript review, the authors have a good opportunity to validate these findings with in vivo experiments. It would be beneficial to see them propose a key and informative next step which they are pursuing along this route of investigation to link with future work.

In addition to discussing future experiments, the authors have the opportunity to propose a PTPN centric mechanism that is driving the disease course and what inhibition of PTPN23 could mean for downstream cell signaling and immune effector responses based off the data presented. It would strengthen the conclusions of this study and

**Do you want your identity to be public for this peer review?** For information about this choice, including consent withdrawal, please see our Privacy Policy

Reviewer #2: **Yes: ** Anthony M. Franchini

---

## [Author Response · Author response to Decision Letter 2]

28 May 2025

Reviewer #2: We thank the authors for their submission, and recognize the hard work they put in preparing this manuscript for submission. It is a thorough study and seeks to leverage a large amount of data in order to identify possible therapeutic targets in osteosarcoma. There are, however, areas which need to be addressed to help the authors obtain their own goal within the current scope of this project to impact their field as they seek to do:

1. While attention has been paid to grammar and editing the English used in this manuscript, it would be worth having a native English speaker take a pass at ending some of the phrases used. Lines such as “The KEGG analysis” on line 196 and use of “based on your findings” on line 200 stand out as easily addressable grammar issues.

Answer Thank you for reviewing our manuscript and for your valuable feedback. We appreciate your attention to the language and grammar aspects of our work. In response to your comments, we have thoroughly revised the entire manuscript for grammar corrections. We have highlighted all changes in red for your convenience. Additionally, we sought input from a native English speaker to ensure the clarity and fluency of our expressions.

2. We appreciate the authors dual focus on T cells and macrophages and the role PTPNs may play in controlling osteosarcoma outcomes in their patient population. To that end, the data focuses on the differentiation from naïve T cells present in the samples into memory and regulatory T cells. However, the authors only use single biomarkers to identify these cell types and do not compare the subgroups across memory and regulatory T cells to see if there is an impact of the tumor microenvironment on their development. A fuller perspective on which T cell populations are affected by the tumor and its impact on cellular PTPN levels would be far informative for the reader. Similar questions may be asked of the monocyte-macrophage aspect of the study, as well.

Answer Thank you for your valuable feedback and for highlighting the importance of our focus on T cells and macrophages in relation to PTPNs and osteosarcoma outcomes.

In response to your concerns regarding the identification of T cell populations and their differentiation, we have conducted further analyses using cell communication methodologies (Cellchat). This approach allowed us to examine the signaling interactions between CD4+ T cells, CD8+ T cells, and monocyte-macrophages, specifically analyzing the ligands and receptors involved in these interactions. Additionally, detailed information on CD4+ T cells (lines 339-349), CD8+ T cells (lines 377-385), and monocyte-macrophages (lines 413-425) is included in the manuscript. The corresponding visualizations can be found in S5 Fig. We believe that these enhancements provide a fuller perspective on how the tumor microenvironment influences the development of these T cell populations and their associated cellular PTPN levels. We appreciate your suggestions and believe that these revisions will make the manuscript more informative and comprehensive for the readers.

3. The authors describe the use of gene enrichment to highlight PTPN family gene significance in active versus inactive gene signatures in Figure 3. However, what is lacking is a clear description of the active versus inactive signatures are, how they were validated in this disease type, and whether the genes identified in this study align with previous reports.

Answer Thank you for your constructive feedback regarding our analysis of PTPN family gene significance in active versus inactive gene signatures.

To differentiate between active and inactive signatures, we performed AUCell analysis on the genes PTPN1 to PTPN23, as illustrated in Fig. 3B. This methodology is detailed in lines 102-103 of the manuscript. Additionally, we conducted pseudobulk analysis along with differential expression and functional enrichment analyses, with results presented in Fig. 3G-H. A thorough description of these analyses and their validation within the context of osteosarcoma is provided in lines 291-295.

Moreover, we discussed the relevance of PTPN23 in relation to other cancers in lines 565-578 of the manuscript. Our findings indicate that PTPN23 has been identified as a tumor suppressor gene in various cancer types, aligning with observations in the existing literature.

4. There is a distinct lack of control in the proliferation studies performed in this study, making it hard to derive validity and impact of the si-PTPN23 transfection into both osteosarcoma cell lines. Is the lack of proliferation due to cell death or simply paused by inhibition of PTPN23? The authors should consider using a non-dye based assay or measure cell viability at these time points to strength the data set.

Answer Thank you for your thoughtful comments regarding our proliferation studies involving si-PTPN23 transfection in osteosarcoma cell lines. In our experiments, we conducted small interfering RNA (siRNA) transfection targeting PTPN23 in osteosarcoma cell lines. To ensure proper controls, we used a transfection reagent for both the treatment group (si-PTPN23) and the control group (si-NC), as clearly indicated in Fig. 8. Additionally, to improve clarity regarding our grouping conditions, we provided detailed descriptions in lines 462-465 of the manuscript.

To further investigate the specific effect of si-PTPN23 on cell proliferation, we performed cell cycle analysis using flow cytometry. Our findings indicated an increase in the S phase after si-PTPN23 treatment, suggesting enhanced cellular proliferation. This is detailed in Supplementary Figure S6, with further discussion provided in lines 466-471 of the manuscript. Regarding your concerns about the perceived lack of cell proliferation, we observed in our cell cycle analysis that the proportion of cells in the S phase increased following si-PTPN23 treatment, indicating that cell proliferation is indeed promoted under conditions of PTPN23 suppression.

5. Echoing the previous manuscript review, the authors have a good opportunity to validate these findings with in vivo experiments. It would be beneficial to see them propose a key and informative next step which they are pursuing along this route of investigation to link with future work.

Answer Thank you for your constructive feedback. I would like to emphasize that we recognize the need for further experiments to deepen our understanding of these findings. We have outlined our plans for additional studies in lines 589-596 of the manuscript.

6. In addition to discussing future experiments, the authors have the opportunity to propose a PTPN centric mechanism that is driving the disease course and what inhibition of PTPN23 could mean for downstream cell signaling and immune effector responses based off the data presented. It would strengthen the conclusions of this study and

Answer Thank you for your insightful feedback and suggestions. In response to your comments regarding the proposal of a PTPN-centric mechanism driving disease progression and the potential therapeutic strategies targeting PTPN23, we have addressed these points in lines 596-604 of the manuscript.

---

## [Editor Report · Decision Letter 2]

Comprehensive Multi-Omics Analysis Reveals the  Prognostic and Immune Regulatory Characteristics of the PTPN Family in Osteosarcoma

PONE-D-24-57357R2

Dear Dr. Liu,

We’re pleased to inform you that your manuscript has been judged scientifically suitable for publication and will be formally accepted for publication once it meets all outstanding technical requirements.

Kind regards,

Zhiwen Luo

Academic Editor

PLOS ONE
---

## [Editor Report · Acceptance letter]

PONE-D-24-57357R2

PLOS ONE

Dear Dr. Liu,

I'm pleased to inform you that your manuscript has been deemed suitable for publication in PLOS ONE. Congratulations! Your manuscript is now being handed over to our production team.

Kind regards,

on behalf of

Dr. Zhiwen Luo

Academic Editor

PLOS ONE